# DNA Vaccines for Epidemic Preparedness: SARS-CoV-2 and Beyond

**DOI:** 10.3390/vaccines11061016

**Published:** 2023-05-23

**Authors:** Joel N. Maslow, Ijoo Kwon, Sagar B. Kudchodkar, Deborah Kane, Amha Tadesse, Hyojin Lee, Young K. Park, Kar Muthumani, Christine C. Roberts

**Affiliations:** 1GeneOne Life Science, Inc., Seoul 04500, Republic of Korea; 2Department of Medicine, Morristown Medical Center, Morristown, NJ 07960, USA

**Keywords:** DNA vaccine, SARS-CoV-2, suction-mediated DNA transfection, vaccine thermostability

## Abstract

We highlight the significant progress in developing DNA vaccines during the SARS-CoV-2 pandemic. Specifically, we provide a comprehensive review of the DNA vaccines that have progressed to Phase 2 testing or beyond, including those that have received authorization for use. DNA vaccines have significant advantages with regard to the rapidity of production, thermostability, safety profile, and cellular immune responses. Based on user needs and cost, we compare the three devices used in the SARS-CoV-2 clinical trials. Of the three devices, the GeneDerm suction device offers numerous benefits, particularly for international vaccination campaigns. As such, DNA vaccines represent a promising option for future pandemics.

## 1. Introduction

The SARS-CoV-2 pandemic highlighted the need for the rapid development of prophylactic vaccines against a new emergent pathogen. This followed a number of recent emergent and re-emergent outbreaks over the past 10 years, including the Ebola virus, Zika virus, the Middle East Respiratory Syndrome coronavirus (MERS-CoV), and rhinovirus, as well as various strains of influenza. These outbreaks of novel pathogens are layered atop both endemic diseases and emerging diseases with increasing geographic spread, such as chikungunya, severe fever with thrombocytopenia syndrome virus (SFTSv), Powassan virus, Nipah virus, and Kyasanur forest disease. While measures such as use of personal protective equipment, social distancing, handwashing, and population lockdowns have yielded some success in limiting transmission, these come at the cost of significant economic and societal strains. For these reasons, the effective long-term prevention of illness and transmission remains reliant on the development and successful rollout of effective vaccines. Critical to global efforts to control future outbreaks is not only the speed of vaccine development but also successfully addressing the logistical challenges of vaccine dissemination, including those confronting developing nations and remote regions.

The SARS-CoV-2 epidemic provided a paradigm for both current and future vaccine development. In addition to “traditional” whole virion-attenuated vaccines, novel vaccine platforms have not only been demonstrated to be effective but constitute a significant proportion of the global vaccination program. Each of these molecular platforms, including mRNA, adenoviral, and DNA vaccines, are often loosely termed as “plug and play” modalities, meaning that synthetic antigenic regions can be easily and quickly designed and introduced into an existing framework. This allowed for rapid clinical advancement, with pivotal Phase 3 studies completed within a year. However, key logistical and cost challenges were quickly identified, which limited the geographic reach of these vaccines. One key factor was the cold-chain requirements for most of these platforms, which necessitated transport and storage at frozen (−20 °C) or ultra-low (−80 °C) temperatures.

In contrast to other vaccine platforms, DNA vaccines have significant thermostability at refrigerated (2–8 °C) and ambient temperatures. These less restrictive requirements regarding cold-chain shipment and storage, especially at the level of individual pharmacies and clinics, provides significant potential advantages for future pandemics and for routine vaccination programs. Thus, DNA vaccines may be ideally suited for regions that are geographically remote as well as for countries and regions where cold-chain requirements pose significant challenges related to cost, a stable supply of electricity, difficulties in overland or air transport, and the distance of clinical sites from storage hubs. DNA vaccines, however, have one unique challenge—the need for a device to ensure in vivo cellular transfection, which can add cost and complexity.

Here, we review DNA vaccines that were developed against SARS-CoV-2 and discuss the potential benefits and challenges of DNA vaccines for future pandemics and for general use.

## 2. DNA Vaccines against SARS-CoV-2

Multiple DNA vaccines against SARS-CoV-2 were developed, with at least 10 advancing into clinical trials [1,2]. While the majority of DNA vaccines represent circular plasmids that express an encoded antigen from an expression vector, other forms, such as linear DNA and minicircle DNA devoid of bacterial elements and antibiotic resistance genes, are also being assessed by different groups with details of the structure and mechanics reviewed elsewhere [2].

The DNA vaccines against SARS-CoV-2 that advanced to Phase 2 clinical testing or beyond are summarized below (Table 1). In this section, we review these vaccines across vaccination schema, safety, and immune responses. Phase 1 studies examined two or more dose levels and/or vaccination schema, of which one regimen was selected to be carried forward into later phases. In each case, the vaccines were well-tolerated without reported vaccine-associated serious adverse effects.

ZyCoV-D, developed by Zydus Pharmaceuticals, has achieved emergency use authorization for general use in India and, thus, is the first DNA vaccine to gain regulatory approval by any regulatory body for human use. ZyCoV-D encodes for the full-length, wild-type spike (S) protein of SARS-CoV-2, which is cloned into the pVAX-1 expression vector. The drug product was formulated in phosphate-buffered saline. A Phase 1/2 study enrolled 48 persons into one of four arms to receive either 1 mg or 2 mg of plasmid administered intradermally (ID) either via needle and syringe injection or using the PharmaJet Tropis^®^ needle-free injection system (NFIS; PharmaJet, Golden, CO, USA) [3]. Vaccinations are carried out on days 0, 28, and 56 for a maximum vaccine dose of 6 mg (Table 1). Immune responses were documented over the 4 weeks post-final vaccination, but long term follow-up was not reported. It was observed that there was a distinct relationship between the dose and humoral responses for the various modes of vaccine administration. Still, this relationship remained largely consistent across the different modes. Seroconversion was documented for the 34.8% study participants who were administered 1 mg of vaccine versus 90% of those administered 2 mg (Table 2). Neutralizing titers trended higher with increasing doses and in vaccines administered with the NFIS (Table 2). T cell responses were similarly dose-dependent but did not vary relative to administration method with maximal responses of approximately 50 SFU/10^6^ cells over baseline (Table 2). The 2 mg dose administered with the NFIS was advanced into Phase 3 studies [4]. The Phase 3 study, conducted in almost 30,000 persons, was performed in India at the time when the Delta variant of SARS-CoV-2 was emergent. Vaccine efficacy for preventing serious illness from SARS-CoV-2 was reported as being 66.6%, which was considered sufficient to gain emergency use authorization in India [4].

INO-4800, developed by Inovio Pharmaceuticals, has been evaluated as part of Phase 1 and Phase 2 clinical trials [5,6]. One Phase 3 study was initially placed on clinical hold and then not restarted after release due to a mismatch between the vaccine, which targeted an ancestral SARS-CoV-2 spike protein, and prevalent strains, which comprised Omicron variants at the time of regulatory go-ahead [7]; however INO-4800 remains as part of a WHO Solidarity Trial. INO-4800 encodes for the wild-type S protein designed as a consensus of available published sequences cloned into the pVAX-1 expression vector. Drug product was formulated in sodium salt citrate (SSC) buffer with vaccine administered via ID injection and was followed by the application of electroporation using the Cellectra 2000^®^ 3P device (Inovio Pharmaceuticals, Plymouth Meeting, PA, USA) [6]. For the Phase 1 study, 40 persons were randomized into two study arms to receive either 1 mg or 2 mg of vaccine at 0 and 4 weeks (Table 1). Immune responses at two weeks post-second vaccination are presented in Table 2 and show a dose-dependent relationship in both the magnitude and frequency of binding antibody and T cell responses relative to the dose administered, whereas neutralizing antibody responses appeared to be dose-independent. Immune responses were reported in the 2 weeks post-vaccination [6]. A dose of 2 mg of plasmid administered at 0 and 4 weeks was advanced to Phase 2 and Phase 3 testing. Preliminary data from the Phase 2 study were posted to a non-peer reviewed preprint server [5]; full study data have not yet been published. One Phase 3 study was terminated prior to completion as noted above; but continues in the Solidarity Trial.

The COVID-eVax developed by Takis Biotech has been evaluated as Phase 1/2 study with Phase 1 data reported [8]. COVID-eVax encodes for the wild-type S protein fused to a tissue plasminogen activator leader sequence that is formulated in phosphate-buffered saline (PBS), stored at −20 °C. For the Phase 1 study, 80 subjects were randomized to receive either 0.5 mg, 1 mg, or 2 mg of the vaccine at an interval of 4 weeks; a fourth group received 2 mg of vaccine only at the baseline. The vaccine was administered intramuscularly (IM) and followed by electroporation using the Cliniporator device (Igea S.p.A., Carpi, Italy). Both antibody and T cell responses trended higher for the 1 and 2 mg dose groups and lowest for the single vaccination group. Both antibody and T cell responses were persistent through 12 weeks but waned by 24 weeks.

AG0302-COVID, developed by AnGes in partnership with Takara, advanced to Phase 2/3, with the results for the Phase 1 study already reported [9]. AG0302 encodes for the SARS-CoV-2 S protein cloned into the pVAX-1 expression vector and formulated in PBS. The Phase 1 study enrolled 20 subjects to receive either 0.2 or 0.4 mg of vaccine at 0 and 2 weeks administered IM via jet injection using the Pyro-Drive Jet Injector (Daicel Corp., Osaka, Japan). Baseline antibody responses were elevated in almost 75% of study participants at study entry, with only four responders in the 0.4 dose group. T cell responses rapidly decreased from a peak of 61.8 two weeks post-vaccination, remaining at approximately 25 SFU/10^6^ cells thereafter. As part of the Phase 2 and Phase 3 studies, subjects were administered 2 mg as a two-dose regimen separated by 2 weeks.

GX-19 and GX-19N were developed by Genexine and advanced to a Phase 2 clinical trials [10]. GX-19 encodes for the SARS-CoV-2 S protein and GX-19N both the S and nucleocapsid (N) proteins. Two Phase 1 studies evaluated two dose levels of GX-19 (1.5 and 3 mg) and 3 mg of GX-19N given as a two-dose regimen separated by 4 weeks by IM injection and followed by EP using the Elimtek device (Seongnam-Si, Republic of Korea). Published data show immediate post-dose responses. Data for GX-19N are shown in Table 2. A Phase 2/3 study was withdrawn prior to recruitment.

GLS-5310, developed by GeneOne Life Science, has been evaluated as part of a combined Phase 1 and 2a clinical trial, with the results of the Phase 1 study recently reported [11]. A pilot trial using GLS-5310 as a booster vaccine in a heterologous regimen with authorized vaccines available to the US public (NCT05182567) is in progress at the time of writing. GLS-5310 encodes for two antigens of SARS-CoV-2: S and ORF3a. Each antigen was designed as a consensus based on published sequences as of February 2020, and both were cloned into a single backbone, pGLS-101, a pVAX-1-based expression vector, to produce a bi-cistronic plasmid vaccine [12]. The drug product was formulated in SSC buffer and the vaccine was administered via ID Mantoux injection followed by the application of suction using the GeneDerm^®^ device (JM-11 model; GeneOne Life Science, Inc., Seoul, Republic of Korea) [13]. In Phase 1 study, 45 persons were allocated into three study arms to receive either 0.6 mg of vaccine administered at 0 and 8 weeks, 1.2 mg administered at 0 and 8 weeks, or 1.2 mg administered at 0 and 12 weeks (Table 1). Immune responses at 4 weeks post-second vaccination appeared to demonstrate differences in immune responses between study arms, but these resolved by the 24-week study visit such that there was no difference relative to dose or timing of vaccinations [11].

Comparisons between these vaccines can only be approximated, since immune responses were determined in separate labs and under distinct assay protocols. Comparing those regimens that were advanced to Phase 2 or beyond show that binding antibody and neutralizing antibody responses would appear to be similar (Table 2). However, T cell responses for GLS-5310 administered with the GeneDerm^®^ suction device appear to be substantially higher than all except for the GX-19N vaccine for which T cell responses of GLS-5310 are approximately two-fold greater than GX-19N. GLS-5310 and GX-19N both encode for two SARS-CoV-2 antigens, although ORF3a and the N antigens account for a small portion, about 20–25%, of the total T cell response [10,11]. The clinical data across different delivery platforms correlate with a preclinical study that compared GLS-5310 administered to rats using either EP, NFIS, or the GeneDerm^®^ suction device [12]. In the latter study, B cell responses were non-distinguishable, whereas T cell responses were also far greater for the vaccine administered with GeneDerm relative to vaccine given using either NFIS or EP [12].

## 3. Persistence of the Immune Response Relative to Delivery Device

The persistence of the immune response following primary vaccination through 48 weeks has been reported for the GLS-5310 vaccine administered with the GeneDerm^®^ suction device [11] and through 24 weeks for the AG0302 vaccine administered with the Pyro-Jet device [9], whereas for other DNA vaccines, data were only reported during the post-vaccination study visits [3,6,8,10]. For GLS-5310, in the CoV2-001 Phase 1 study, binding antibody responses remained stable over the year of follow-up with binding antibody GMT in the 700 to 1000 range. T cell responses increased through study week 24 to approximately 1200 SFU/10^6^ cells and then remained stable through Week 48 [11]. For AG0302, T cell responses remained at approximately 25 SFU/10^6^ cells through 6 months for recipients of the 0.4 mg dose; B cell responses remained low for both dose groups [9].

## 4. Booster Vaccination

GLS-5310 is currently being assessed as a part of a heterologous booster vaccine for those previously administered either the BNT162b2 (Pfizer, NY, NY, USA) or mRNA-1273 (Moderna, Cambridge, MA, USA) RNA vaccines or the Ad26.CO2.S (Janssen, Beerse, Belgium) adenoviral vaccine at least 6 months prior to GLS-5310 administration (NCT05182567). The study is ongoing as of this writing.

As part of the Phase 1 study of GLS-5310, 32 of the 45 participants elected to receive one of the two authorized mRNA vaccines during the extended follow-up period. As has been presented at the 2022 Congress of the International Society of Vaccines [14], the receipt of an mRNA vaccine approximately 8 months after GLS-5310 DNA vaccination yielded a greater than 100-fold increase in neutralizing antibody titers, a greater than 1000-fold increase in binding antibody titers, and a 3-fold increase in T cell responses, the latter from a pre-mRNA baseline of about 1200 SFU/10^6^ cells. In two reported studies of heterologous boost vaccination, post-boost immune responses were similar, regardless of which vaccine type was used for the primary vaccine series and which was used as the booster (e.g., mRNA primary vaccination followed by an Ad26 boost versus Ad26 primary vaccination followed by an mRNA boost) [15,16]. This study and our aforementioned booster study will provide key information about the utility of DNA vaccines as part of a booster regimen.

## 5. Dose Response

As detailed in Table 2, immune responses following vaccination with either the ZyCoV-D or the INO-4800 vaccines appeared to be dose-dependent. However, while there were apparent differences in immune responses immediately post-vaccination for GLS-5310, these were non-significant, as immune responses for some dose groups appeared to peak later such that group differences waned over time as immune responses matured. Whether apparent dose level differences would have similarly resolved for the ZyCoV-D or INO-4800 vaccines is unknown, as long-term follow-up data were not reported. However, for both the ZyCoV-D and INO-4800 vaccines, preclinical studies showed no evidence of a dose response for either B or T cell responses across 4-fold differences in the administered dose of vaccine [17,18]. Reported data also show a correlative trend between dose and immune responses for the COVID-eVax, AG0302, and GX-19 vaccines [8,9,10].

The finding that immune responses following vaccination with GLS-5310 were dose-independent is similar to the results of our prior clinical and preclinical studies. Pre-clinically, we found that both antibody and T cell responses were not significantly different for rats administered either 30 or 300 µg of GLS-5310 [12]. In a prior study of the GLS-5700 vaccine against the Zika virus, built on the same DNA vaccine platform, participants vaccinated with either 1 mg or 2 mg developed immune responses that were not statistically different [19]. Additionally, in a study of another of the same DNA vaccine platform modality, the GLS-5300 vaccine against the MERS coronavirus, participants vaccinated with either 0.6, 2, or 6 mg of vaccine at 0, 4, and 12 weeks similarly displayed no significant difference in immune response [20].

## 6. Vaccine Efficacy

Of the vaccines against SARS-CoV-2, only one, ZyCoV-D, completed Phase 3 and had a reported efficacy to prevent serious illness of 67% [4]. While less than the 95% reported efficacy for the two approved mRNA vaccines, it is similar to the 60–70% reported efficacy for the adenoviral vaccines [21,22].

## 7. Devices

For the three SARS-CoV-2 DNA vaccines that advanced into clinical trials, each utilized a different device to enhance in vivo transfection. In all three cases, the DNA was injected “naked” into the intradermal space, i.e., without incorporation into a nanoparticle in the formulation. Two of the device technologies, jet droplet delivery and suction, can be broadly classified as mechano-transfection devices and the third as an electrotransfection or electroporation device.

A number of electroporation devices have been used in clinical trials, including Cellectra^®^, Trigrid^®^ (Ichhor Medical Systems, San Diego, CA, USA) Elimtek device, and Cliniporator^®^ [23,24,25]. Following either intradermal or intramuscular injection of DNA, an electric field is generated across electrodes inserted into the tissue just external to the region where the DNA has been injected. Different devices utilize different pulsing parameters and may apply either a fixed current or fixed voltage across the electrodes with a typical applied voltage of 200 V. Electroporation is known to cause tissue trauma that has been postulated as being integral to the generation of the vaccine’s immune response [26]. Tissue trauma is also evident clinically, and increased levels of creatinine kinase have been reported following EP with intramuscular devices [20], and injection site scabbing with resultant scarring or cheloid formation has been observed with use of intradermal EP devices [19].

The needle-free injection system (NFIS) is a mechanical device that injects microdroplets into skin or muscle. After the device is placed against the skin, a spring-loaded actuator forces liquid formulated vaccine through a microaperture, creating the microdroplets. The NFIS has been used to deliver both protein and DNA vaccines, including a Zika DNA vaccine [27] and a DNA vaccine against SARS-CoV-2 [3,4]. Other than minimal to moderate pain during injection, there is no evident tissue trauma associated with use of the NFIS.

Our group utilized the GeneDerm^®^ suction device as part of the clinical trials of the GLS-5310 SARS-CoV-2 vaccine [11]. Following the injection of vaccine intradermally, the GeneDerm^®^ device is applied directly over the injection site. The device used in the study generated a suction pressure of 65 kPa for 30 s. Preclinical studies showed that DNA transgene expression following suction-induced transfection was detectable within an hour [13]. The enhancement of cellular DNA uptake with suction was evident for rats administered a plasmid expressing green-fluorescent protein (GFP) and then confirmed for rats vaccinated with the SARS-CoV-2 DNA vaccine with or without the application of suction. Immune responses for rats administered vaccine with GeneDerm^®^ suction were over 100 times greater than injection alone, and immune responses were dose-independent for rats vaccinated with either 30 µg or 300 µg of DNA [13]. Importantly, suction did not cause trauma to skin tissue when investigated either in visual and/or histologic examination [13], a finding that was corroborated in a GLP toxicology study [28]. A preclinical study compared the GLS-5310 vaccine delivered with GeneDerm, EP, and NFIS and showed that B cell responses were not significantly different, whereas T cell responses were greatest with suction mediated DNA delivery [12]. GeneDerm has now been used in three clinical trials of the GLS-5310 DNA vaccine with a very favorable use profile.

Table 3 provides a comparison of the three devices based on a number of relevant user parameters and cost considerations. GeneDerm^®^ is the most intuitive of the three devices in its use, although the NFIS is a close second. The use of GeneDerm^®^ device is being targeted not only for more advanced countries and regions, but its simplicity and portability also make it ideal for use in remote and resource-poor regions, where hands-on training will not be available and cost is a limiting factor. With regard to price, the GeneDerm^®^ suction device is orders of magnitude less expensive than either NFIS or EP, thus adding little additional expense to the total vaccination cost. Finally, disposables for the three devices should strongly favor GeneDerm^®^. GeneDerm^®^ only requires the replacement of an inexpensive cap atop the device tip, whereas the NFIS requires a new syringe with each use and EP devices require the replacement of complex electrode arrays. Projected storage and shipment costs and volume requirements for storage similarly mimic the order of GeneDerm^®^ to NFIS to EP from least to greatest.

## 8. Advantages of DNA Vaccines

DNA vaccines have a number of advantages over other vaccine platforms with regard to manufacture, stability, logistics, toxicology, and T cell responses.

The manufacture of DNA vaccines under cGMP is available from a number of contract development manufacturing organizations (CDMO), with only a few CDMOs having the capability and capacity to manufacture at scale. Companies with large-scale abilities include VGXI, Inc., which has recently opened a new 125,000 SF facility in Conroe, TX USA, for commercial scale production, complementing its original 40,000 facility in the Woodlands, TX, USA, and Richter-Helm in Hamburg, Germany. Both companies offer high-purity DNA at high levels of supercoiling (>95%) and can manufacture at concentrations up to 10 mg/mL. VGXI has demonstrated the ability to respond rapidly during epidemics—including Ebola [29], MERS-CoV [20], Zika [19], and SARS-CoV-2 [6,11]. Moreover, the time from concept to the manufacture of cGMP plasmid included the creation of a cell bank in *E. coli*, process development testing, large-scale fermentation and purification, and the fill/finish of the vaccine was accomplished in as little as 6 weeks. The efficiency of manufacture allowed a Zika vaccine to progress from concept to Phase 1 clinical trial in 7 months [19], which was then the shortest time to clinical testing for any vaccine, and a SARS-CoV-2 vaccine advanced from concept to first participant dosing in 2.5 months [6].

Perhaps the single greatest advantage regarding the manufacturing and usage of DNA and DNA vaccines is its thermostability. Most vaccine platforms have significant cold-chain requirements for transport and storage at −20 °C or even −80 °C. We routinely store DNA vaccines under refrigerated conditions (+2–8 °C) and have found these to be stable for up to 3 years. An anecdotal report additionally noted DNA vaccine stability at +25 °C for 1 year and at 37 °C for a month [29]. A study of accelerated degradation conditions comparing structural stability and Immunogenicity for the COVIDe-Vax SARS-CoV-2 vaccine stored frozen at −20 °C versus vaccines kept at 23 °C, 45 °C, and 65 °C [30]. Whereas exposure to 45 °C caused the almost complete conversion from supercoiled to open circle forms, gene expression and binding antibody and T cell responses were unaffected in contrast to storage at 65 °C, which resulted in the linearization of the DNA and a significant reduction in gene expression, resulting in the almost complete loss of T cell responses and decreased B cell responders [30]. Our group is conducting a 2-year long-term accelerated stability study of the GLS-5310 SARS-CoV-2 vaccine and have similarly noted no loss of immunogenicity for vaccines held at 25 °C for a year, 32.5 °C for 6 months, and 37 °C for 3 months (unpublished data). The minimal requirements for DNA vaccine cold-chain storage and transport both significantly decrease vaccine cost and dramatically increase the geographic reach of any vaccination program. Moreover, stability at both ambient (25 °C) and higher temperatures (to 37 °C) increases the potential success of field programs.

DNA vaccines appear to avoid platform-specific adverse effects that have been observed with mRNA and adenoviral vaccines. The mRNA vaccines are associated with significant reactogenicity and allergic reactions, including anaphylaxis, which may be related to the polyethylene glycol (PEG) component of the LNP [31,32,33]. Additionally, whereas the formation of anti-PEG antibodies following vaccination with an mRNA SARS-CoV-2 vaccine may not be common [34], there is a report of the formation of anti-PEG antibodies in two patients that resulted in the clearance of PEG-complexed recombinant human Factor VIII and the subsequent loss of anticoagulant activity attributed to recent mRNA vaccination [35]. The adenoviral vaccines appear to have a risk for the development of unusual thrombotic complications, including central venous thrombosis [36].

Finally, DNA vaccines have, in general, induced superior T cell immune responses. For SARS-CoV-2, the magnitude of T cell responses has directly correlated with superior outcomes [37,38,39,40]. For SARS-CoV-2 DNA vaccines, it is of note that the T cell responses of the mRNA, adenoviral, subunit, and two of the DNA vaccines were similar (approximately 50 SFU/10^6^ PBMCs) [15,16,41], whereas reported T cell responses for GLS-5310 administered with the GeneDerm^®^ suction device were approximately 1200 SFU/10^6^ PBMCs [11]. All three DNA vaccines had a high rate of responders post-vaccination. Prior studies of DNA vaccines, as platform technologies with similar backbones and formulations, showed that the Zika DNA vaccine induced T cell responses in the range of 125–175 SFU/10^6^ PBMCs, which remained stable for 36 weeks [19]. A MERS-CoV DNA vaccine induced T cell responses in approximately 70% of recipients in the range of 300–700 SFU/10^6^ PBMCs that persisted for 60 weeks [20].

## 9. Challenges of DNA Vaccines

In terms of widespread usage, DNA vaccines have two particular challenges. As noted in this review, it is essential to note that DNA vaccines necessitate a device to enhance cellular uptake, which, in turn, promotes maximal protein expression. This adds to the cost of vaccination programs and can increase the complexity of delivery. Only the PharmaJet NFIS has been part of a vaccine/device combination approved for use, whereas EP and the GeneDerm^®^ suction device are still considered experimental.

A second challenge faced by DNA vaccines for SARS-CoV-2 is the relatively lower antibody and neutralizing antibody responses compared to other vaccine platforms. Specifically, the reported responses were approximately 2-log lower than those observed for mRNA vaccines and about 1-log lower than those induced by adenoviral vaccines. Whether this explains the differential efficacy for the vaccine types is unknown, since the studies were conducted in different countries and during different stages of the pandemic when differing variants were dominant: the wild-type virus was dominant at the time when studies on the mRNA vaccine were ongoing, whereas the Delta variant was likely dominant at the time of the testing of the ZyCoV-D vaccine in India. It is important to note that no correlation of protection has been identified thus far that can link the response to any DNA vaccines with efficacy through laboratory measures. Multiple research groups have studied various strategies to improve DNA vaccine-binding antibody and neutralizing antibody responses. Some of these approaches involve using molecular adjuvants, which may include co-administered DNA-encoded chemokines, cytokines, and other molecules, such as perforin and lipid nanoparticles. Further details on these approaches are available in other relevant publications [42,43,44].

## 10. Conclusions

DNA vaccines possess several attributes that make them suitable for future vaccine campaigns. One notable benefit is the minimal cold-chain requirements for transportation and storage and their stability at elevated temperatures when deployed in the field. Although the experience with DNA vaccines may not match that of viral-based delivery or mRNA-based vaccines, they have demonstrated favorable safety profiles. In addition, researchers are actively investigating ways to enhance the immune response of DNA vaccines by using molecular adjuvants.

## Figures and Tables

**Table 1 vaccines-11-01016-t001:** DNA vaccines against SARS-CoV-2—schema chosen for development.

Vaccine	Developer	Development Phase	Dose per Vaccination	Dosing Regimen	Delivery Device	Mechanism
ZyCoV-D	Zydus	Authorized (India)	2 mg	0–4–8 weeks	PharmaJet^®^	NFIS
INO-4800	Inovio	Phase 3	2 mg	0–4 weeks	Cellectra^®^	EP
COVID-eVax	Takis	Phase 2	2 mg	0–4 weeks	Cliniporator^®^	EP
AG0302	AnGes	Phase 3	0.4 mg	0–2 weeks	Pyro-drive jet injector	NFIS
GX-19N	Genexine	Phase 2	3 mg	0–4 weeks	Elimtek EP	EP
GLS-5310	GeneOne	Phase 2	1.2 mg	0–8 weeks	GeneDerm^®^	Suction

**Table 2 vaccines-11-01016-t002:** Immune responses immediately post-vaccination.

	ZyCoV-D				INO-4800		COVID-eVax	AG0302	GX-19N	GLS-5310		
Dose (mg) > Device > Schema (wks) >	1 mg Inj 0–4–8	1 mg NFIS 0–4–8	2 mg Inj 0–4–8	2 mg NFIS 0–4–8	1 mg 3P 0–4	2 mg 3P 0–4	2 mg ClinEP 0–4	0.4 mg Pyro-jet 0–2	3.0 mg Elimtek 0–4	0.6 mg GD 0–8	1.2 mg GD 0–8	1.2 mg GD 0–12
Binding antibody end point titer	39	96	748	884	331.2	691.4	6 IU/mL	324.9	201.6	708.0	645.5	241.4
Responder	36%	33%	100%	80%	74%	100%	80%	NR	75%	93%	93%	93%
Neutralizing antibody end point titer	8.5	12.0	14.1	39.2	44.4	34.9	10	7.6	24	25	15	25
Responder	18%	17%	50%	80%	78%	84%	10%	NR	100%	33%	73%	60%
SFU/10^6^ cells	10	70	16	64	26.2	71.1	150	61.8	750	576	1090	482
Responder	100%	100%	100%	100%	74%	100%	90%	NR	52%	93%	93%	93%

For ZyCoV-D, values are shown on study day 84, 1 month post-vaccination. T cell responses for the 1 mg injection-only group for ZyCoV-D were ~120 and 64 SFU/10^6^ cells for the 1 mg NFIS group on day 56, whereas for other groups, maximal responses were observed at day 84. Baseline values for binding antibodies and neutralizing antibodies respectively, averaged 14.8 and 6.3 across groups. Baseline T cell responses averaged 11 SFU/10^6^ cells. For INO-4800, values are shown for week 6, 2 weeks post-vaccination. Baseline titers for binding and neutralizing antibodies were 0 and 16, respectively. For COVD-eVax, values are shown for week 8, 4 weeks post-vaccination. Baseline antibody responses were not detected; baseline T cell responses averaged 50 SFU/10^6^ cells. For AG0302, values are shown for week 4, 2 weeks post-vaccination. Baseline binding antibody responses were elevated with a titer of 324.9, whereas neutralizing antibody responses at baseline were 1.6 and T cell responses of 6 SFU/10^6^ cells. For GX-19N, values are shown for day 57, 4 weeks post-vaccination for antibody responses and day 43 for T cell responses. Binding antibody responses at baseline were 24.4 and 20 for binding antibody and neutralizing antibody responses and 200 for T cell responses. For GLS-5310, values are shown for 4 weeks post-second vaccination. Baseline titers for binding and neutralization antibodies were non-reactive. Baseline values for T cell responses averaging 67 SFU/10^6^ cells. Abbreviations: Inj, injection; NFIS, PharmaJet Tropis^®^ needle-free injection system; 3P, Cellectra 2000-3P^®^ electroporation device; ClinEP, Cliniporator electroporation device; GD, GeneDerm^®^ suction device, NR, not reported.

**Table 3 vaccines-11-01016-t003:** Relative comparison of intradermal in vivo DNA transfection devices.

	EP	NFIS	Suction
User considerations			
Training needs	High	Medium	Low
Intuitive design	Low	Low	High
User feedback for successful use	Medium	Low	High
Ability to reapply device if error	No	No	Yes
Pain, discomfort	High	Medium	Low
Time of application and readiness for next use			
Recharging requirement	Yes	No	Yes
Cost considerations			
Device cost	High	Medium	Low
Disposables cost per use (estimated)	$5	~$1	$0.25

## Data Availability

Not. applicable.

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
