# Peer review of "DNA Vaccines for Epidemic Preparedness: SARS-CoV-2 and Beyond"

_vaccines, 2023, doi:10.3390/vaccines11061016_

Round 1

Reviewer 1 Report

The revise article seems to be ok.

Author Response

We thank Reviewer 1 for the time to review the paper. 

Reviewer 1 had no comments. 

Reviewer 2 Report

This is very nice review paper to indicate important and advantage of DNA vaccine for epidemic preparedness. There are only minor comments.

1.       In page 3: “INO-4800, developed by Inovio Pharmaceuticals, has been evaluated as part of Phase 1 and Phase 2 clinical trials [3, 4]. A Phase 3 study was terminated early and not completed [5].” The authors may add the possible reasons why the phase 3 study was not completed.

2.       In page 4: “Of note is that T cell responses continued to increase through study week 24 to approximately 1200 SFU/106 cells and remained consistent through Week 48 [6].” Check this sentence.

(Of note is that T cell responses continued to increase to approximately 1200 SFU/106 cells at week 24 of the study and remained that level consistently  through Week 48 [6].)?

No concern

Author Response

We thank Reviewer 2 for their careful reading and critique of the manuscript. The comments and responses are below. 

This is very nice review paper to indicate important and advantage of DNA vaccine for epidemic preparedness. There are only minor comments.

  1. In page 3: “INO-4800, developed by Inovio Pharmaceuticals, has been evaluated as part of Phase 1 and Phase 2 clinical trials [3, 4]. A Phase 3 study was terminated early and not completed [5].” The authors may add the possible reasons why the phase 3 study was not completed.

Response: The reasons for the termination of the study has been added to the paper.

  1. In page 4: “Of note is that T cell responses continued to increase through study week 24 to approximately 1200 SFU/106 cells and remained consistent through Week 48 [6].” Check this sentence.

(Of note is that T cell responses continued to increase to approximately 1200 SFU/106 cells at week 24 of the study and remained that level consistently  through Week 48 [6].)?

Response: This sentence has been reworded

Reviewer 3 Report

This article offers an overview of DNA vaccines used in the era of COVID-19. Three DNA vaccines are described with delivery routes, immunogenicity and dosage. Lastly, the advantages and challenges of DNA vaccines over other vaccines are highlighted.

To mu knowledge, all these informations were reviewed by Shafaati et al., 2022 in the Future Medicine Ltd, Future Virology Volume 17, Issue 1, January 2022, Pages 49-66. https://doi.org/10.2217/fvl-2021-0170.

Therefore, I miss the novelty of the present literature review. It would be better to wait the results of the ongoing clinical trials and book a complete review.

Author Response

We thank Reviewer #3 for their review and comments. The critique and response are below. 

This article offers an overview of DNA vaccines used in the era of COVID-19. Three DNA vaccines are described with delivery routes, immunogenicity and dosage. Lastly, the advantages and challenges of DNA vaccines over other vaccines are highlighted.

To mu knowledge, all these informations were reviewed by Shafaati et al., 2022 in the Future Medicine Ltd, Future Virology Volume 17, Issue 1, January 2022, Pages 49-66. https://doi.org/10.2217/fvl-2021-0170.

Therefore, I miss the novelty of the present literature review. It would be better to wait the results of the ongoing clinical trials and book a complete review.

Response: We thank the reviewer for their comments. The review of DNA vaccines from Shafaati and another review from Chavda have been added to this review. Both papers are complementary to the information presented in this manuscript. Shaafati and Chavda both provide details as to the various forms that DNA vaccines may take, as well their mechanism of action. Since the publication of the two prior reviews, results from most of the phase 1 studies have since been published and are reviewed and compared here. We have additionally added details of the other DNA vaccines for which immunogenicity data has been published.

This paper additionally addresses in detail the issue of thermostability of DNA vaccines as well as a more fulsome comparison of the various devices used in conjunction with DNA vaccines to induce in vivo transfection.

Round 2

Reviewer 3 Report

Dear authors

Thank you very much for updating and inserting the comments highlighted.